# Accuracy of Fibrosis-4 Index in Identification of Patients with Cirrhosis Who Could Potentially Avoid Variceal Screening Endoscopy

**DOI:** 10.3390/jcm9113510

**Published:** 2020-10-29

**Authors:** Koji Ishida, Tadashi Namisaki, Koji Murata, Yuki Fujimoto, Souichi Takeda, Masahide Enomoto, Hiroyuki Ogawa, Hirotetsu Takagi, Yuki Tsuji, Daisuke Kaya, Yukihisa Fujinaga, Masanori Furukawa, Yasuhiko Sawada, Kou Kitagawa, Shinya Sato, Norihisa Nishimura, Hiroaki Takaya, Kosuke Kaji, Naotaka Shimozato, Hideto Kawaratani, Kei Moriya, Takemi Akahane, Akira Mitoro, Hitoshi Yoshiji

**Affiliations:** Department of Gastroenterology, Nara Medical University, 840 Shijo-cho, Kashihara, Nara 634-8522, Japan; ishidak@naramed-u.ac.jp (K.I.); muratak@naramed-u.ac.jp (K.M.); yukifuji@naramed-u.ac.jp (Y.F.); souitit@naramed-u.ac.jp (S.T.); masahidee@naramed-u.ac.jp (M.E.); ogawah@naramed-u.ac.jp (H.O.); htakagi@naramed-u.ac.jp (H.T.); tsujih@naramed-u.ac.jp (Y.T.); kayad@naramed-u.ac.jp (D.K.); fujinaga@naramed-u.ac.jp (Y.F.); furukawa@naramed-u.ac.jp (M.F.); yasuhiko@naramed-u.ac.jp (Y.S.); kitagawa@naramed-u.ac.jp (K.K.); shinyasato@naramed-u.ac.jp (S.S.); nishimuran@naramed-u.ac.jp (N.N.); htky@naramed-u.ac.jp (H.T.); kajik@naramed-u.ac.jp (K.K.); shimozato@naramed-u.ac.jp (N.S.); kawara@naramed-u.ac.jp (H.K.); moriyak@naramed-u.ac.jp (K.M.); stakemi@naramed-u.ac.jp (T.A.); mitoroak@naramed-u.ac.jp (A.M.); yoshijih@naramed-u.ac.jp (H.Y.)

**Keywords:** FIB-4, Baveno VI consensus, esophageal varices, liver cirrhosis, serum fibrosis index

## Abstract

A potential restriction of the Baveno VI consensus, which helps to avoid unnecessary endoscopies, is the limited availability of FibroScan. We aimed to identify serum fibrosis indices that might aid in ruling out the presence of high-risk varices in cirrhotic patients. This retrospective study included 541 consecutive patients with cirrhosis who underwent endoscopy and had data available for nine serum fibrosis indices, including platelet count, hyaluronic acid, 7S fragment of type 4 collagen, procollagen type III N-terminal peptide, tissue inhibitor of metalloproteinases 1, Mac-2 binding protein glycosylation isomer, fibrosis index based on four factors (FIB-4), aspartate transaminase/platelet ratio index and enhanced liver fibrosis score. Optimal index cutoffs for predicting high-risk varices were calculated in an estimation cohort (*n* = 127) and evaluated in a validation cohort (*n* = 351). The diagnostic performance of the indices was assessed by receiver operating characteristic curve analysis. In the estimation cohort, a FIB-4 cutoff of 2.78 provided the greatest diagnostic accuracy in predicting both all-grade and high-risk varices. FIB-4 had a negative predictive value of 1.00 for high-risk varices in both cohorts, and 21.3% (27/127) and 14.8% (52/351) of the estimation and validation cohorts, respectively, avoided esophagogastroduodenoscopy; no high-risk varices were missed in either cohort. FIB-4 correctly identifies the absence of high-risk varices in patients with cirrhosis. Therefore, those with a FIB-4 of ≥2.78 should undergo esophagogastroduodenoscopy, and FIB-4 determination should be recommended every 6–12 months concurrently with the other blood tests until the index value reaches 2.78 in those with a FIB-4 of <2.78.

## 1. Introduction

Development of portal hypertension, a hallmark of disease progression or poor prognosis in the clinical course of liver cirrhosis, can consequently lead to the aggravation of esophageal varices and worsening ascites [1]. Esophagogastroduodenoscopy, the gold standard method for the diagnosis of gastroesophageal varices, is recommended for variceal screening at the time of diagnosis in all patients with cirrhosis; however, the procedure is costly and may be unpleasant [2]. Universal screening endoscopy, which has been shown to be beneficial in certain patients, is invasive for routine screening and follow-up assessment of esophageal varices [3]. The Baveno VI consensus workshop proposes that patients with compensated advanced chronic liver disease accompanied with a liver stiffness < 20 kPa and a platelet count > 150,000/mm^3^ are at very low risk for high-risk varices and could potentially avoid screening endoscopy [4,5]. Although measurement of the hepatic venous pressure gradient is the gold standard technique for the evaluation of portal hypertension in chronic liver disease, transient elastography (FibroScan^®^) represents the most promising noninvasive technique that may substitute this invasive evaluation method for variceal detection [6,7]. However, transient elastography is not widely available and is primarily used at academic institutions due to high cost [8].

Progression of liver fibrosis is a major cause for the development of portal hypertension in liver cirrhosis. No serum markers have been reported to accurately reflect portal hypertension [9]. Serum liver fibrosis indices exhibit modest diagnostic performance in predicting the presence of varices in patients with liver cirrhosis [10,11]. Conversely, a recent systematic review and meta-analysis of randomized controlled trials has revealed that several fibrosis indices are inadequate to replace variceal screening endoscopy [12]. The fibrosis index based on four factors (FIB-4) is a useful noninvasive tool for estimating the severity of fibrosis in patients with various chronic liver diseases [13,14]. The diagnostic performance of FIB-4 for advanced liver fibrosis in nonalcoholic fatty liver disease has been reported in several studies [15,16]. In clinical practice, a low FIB-4 cutoff of 1.45 represents an easily accessible tool to rule out patients without advanced fibrosis [17]. Shah et al. have shown that a FIB-4 cutoff of 1.3 can identify patients without advanced fibrosis [18], illustrating the utility of FIB-4 as a triaging test. In the present study, we compared the diagnostic performance of noninvasive serum fibrosis indices in predicting the presence of esophageal varices as well as high-risk varices in patients with cirrhosis, with the aim to identify serum indices that could identify patients with cirrhosis who might avoid futile screening endoscopy.

## 2. Patients and Methods

### 2.1. Patients and Study Protocol

This was a retrospective observational study conducted at the Nara University Hospital in Nara, Japan. Patients who were diagnosed with liver cirrhosis based on laboratory and imaging tests or liver biopsy from 1 January 2016 until 31 December 2019 were enrolled. Data, including the results of esophagogastroduodenoscopy for varices and laboratory tests, were retrospectively analyzed. Patients with hepatocellular carcinoma, those taking beta-blockers or nitrates, those with portosystemic shunting, those with a history of endoscopic treatment for esophageal varices, gastrointestinal surgery, gastrointestinal malignancies other than HCC, or thrombosis of the portal and splenic vein were excluded. Among a total of 547 patients in the study period (Figure 1), 478 patients who underwent esophagogastroduodenoscopy within one year prior to enrollment were included in the present study.

The esophageal varices were categorized as follows based on their form: F1, straight and small; F2, moderately enlarged and beady and F3, markedly enlarged and nodular. F1 esophageal varices were defined as low risk, whereas F2 and F3 esophageal varices were defined as high-risk, based on 2016 practice guidance by the American Association for the study of liver diseases [19].

Informed consent was obtained from all participants before the study initiation. This study was approved by the Human Ethics Review Committee of Nara Medical University Hospital and was conducted in accordance with the Declaration of Helsinki.

### 2.2. Laboratory Data

The collected laboratory data included serum levels of aspartate aminotransferase (AST), alanine aminotransferase, albumin, total bilirubin, prothrombin time and the following noninvasive indices: platelet counts (PLT) and serum levels of hyaluronic acid, 7S fragment of type 4 collagen (7S collagen), procollagen type III N-terminal peptide (PIIINP), tissue inhibitor of metalloproteinases-1 (TIMP-1) and Mac-2-binding protein glycosylation isomer (M2BPGi); FIB-4; AST/platelet ratio index (APRI) and enhanced liver fibrosis score based on the combination of TIMP-1, PIIINP and hyaluronic acid values [10,16,20].

### 2.3. Assessment of Varices and Bleeding Risk

All endoscopic procedures were performed by gastroenterologists. All patients underwent esophagogastroduodenoscopy within one year of the laboratory tests. Optimal cutoff values to exclude the presence of high-risk varices were estimated for nine noninvasive fibrosis indices in an estimation cohort, which comprised 127 patients with cirrhosis who underwent esophagogastroduodenoscopy and had data available for all nine indices. The determined cutoff values were used in a validation cohort, which comprised 351 patients with cirrhosis and available FIB-4 values who underwent esophagogastroduodenoscopy. Sensitivity, specificity, positive predictive value (PPV), negative predictive value (NPV) and accuracy were calculated for all fibrosis indices evaluated in the study.

### 2.4. Statistical Analysis

Continuous variables were presented as median (50th percentile) and interquartile range (25th and 75th percentiles). All statistical analyses were performed using the SPSS statistical software package version 22.0 (IBM, Armonk (HQ), NY, USA). The diagnostic accuracy of the serum fibrosis indices (sensitivity, specificity, PPV, NPV and accuracy) was determined using the area under receiver operating characteristic curves. *p* values < 0.05 were considered to indicate statistical significance.

## 3. Results

### 3.1. Demographic Characteristics

Of the study cohort of 478 patients with cirrhosis, 127 patients (82 males (64.6%) and 45 females (35.4%)) were included in the estimation cohort, whereas the remaining 351 patients (217 males (61.8%) and 134 females (38.2%)) were included in the validation cohort.

The serum albumin levels were significantly lower in the validation cohort than in the estimation cohort. However, no significant differences were observed in sex, etiology of liver cirrhosis, Child–Pugh score, serum levels of AST, alanine transaminase, total bilirubin, prothrombin time, platelet count, Mayo end-stage liver disease score, FIB-4 or APRI (Table 1). Patients with cirrhosis in the validation cohort developed high-risk varices significantly more often than those in the estimation cohort. FIB-4 was developed using four simple parameters, including age, which might affect the diagnostic performance of FIB-4 in predicting portal hypertension or the presence of varices [21]. Of note, there was no difference in age between the two cohorts.

### 3.2. Performance of a FIB-4 Cutoff of 2.78 in the Estimation Cohort

The estimation cohort comprised patients with cirrhosis who underwent esophagogastroduodenoscopy and had data available for all nine serum liver fibrosis indices (*n* = 127). The optimal cutoff values for all nine serum fibrosis indices were calculated using receiver operating characteristic curve analysis to identify the presence of all-grade varices and high-risk varices. Considering that the cutoff values were used to predict the presence of all-grade varices in patients with cirrhosis, we aimed to choose cutoff values that achieved maximum sensitivity rather than maximum specificity. As a result, the individual cutoff values for fibrosis markers to predict the presence of high-risk varices and their sensitivity, specificity, PPV and NPV in the estimation cohort are presented in Table 2. Specifically, the cutoff values for FIB-4, hyaluronic acid, platelet count, enhanced liver fibrosis score, 7S collagen, APRI, M2BPGi, PIIINP and TIMP-1 were 2.78, 110.63 ng/mL, 11.9 × 10^3^/μL, 11.75, 6.1 ng/mL, 0.89, 1.47 cutoff index (C.O.I.), 0.6 ng/mL and 379.9 ng/mL, respectively. In the estimation cohort, a FIB-4 cutoff of 2.78 provided the greatest diagnostic accuracy (area under the receiver operating characteristics (AUROC) = 0.69) to predict all-grade varices. FIB-4 showed an NPV of 100% in detecting high-risk varices in the estimation cohort. Additionally, 68 of the 100 patients with FIB-4 values ≥ 2.78 (68%) had all-grade varices, including 11 (11%) patients with high-risk varices. Conversely, 7 of the 27 patients with FIB-4 values < 2.78 (25.9%) had all-grade varices, none of which were high-risk (Figure 2).

### 3.3. Performance of a FIB-4 Cutoff of 2.78 in the Validation Cohort

In the validation cohort (*n* = 351), the sensitivity, specificity, PPV and NPV were 100%, 20%, 29% and 100%, respectively, for the recommended FIB-4 cutoff of 2.78 to predict high-risk varices (Table 3). Additionally, among the patients with FIB-4 values ≥ 2.78, 189 (63.2%) had all-grade varices, and 88 (29.4%) had high-risk varices. Furthermore, 14 of the 52 patients with FIB-4 values < 2.78 (26.9%) had all-grade varices, none of which were high-risk varices (Figure 3).

### 3.4. Performance of a FIB-4 Cutoff of 3.2 in the Estimation and Validation Cohort

Our analyses to evaluate the performance of the recommended optimal FIB-4 cutoff value for high-risk varices in the estimation cohort are shown in Appendix A. In the validation cohort (*n* = 351), the sensitivity, specificity, PPV and NPV were 94%, 25%, 30% and 93%, respectively, using the recommended FIB-4 cutoff of 3.2 to determine high-risk varices (Appendix A). In the estimation cohort (*n* = 127), 63 of the 94 patients (67.0%) with FIB-4 values ≥3.20 had all-grade varices; of these, 11 (11.7%) were high-risk varices. Conversely, 12 of the 33 patients (36.4%) with FIB-4 values <3.20 had all-grade varices, none of which were high-risk varices (Appendix A). In the validation cohort, 179 of the 280 patients (63.9%) with FIB-4 values ≥3.20 had any varices; of these, 83 (29.6%) were high-risk varices. Conversely, 25 of the 71 patients (35.2%) with FIB-4 values < 3.20 had all-grade varices. Finally, five of the 71 (7.0%) of high-risk varices were missed using a FIB-4 cutoff of 3.20 (Appendix A).

### 3.5. Performance of a Combination of FIB-4 and the Other Fibrosis Indices

The diagnostic performances of a combination of FIB-4 and the other fibrosis indices were evaluated using the ROC curves. The individual cutoff values for FIB-4 alone and those for a combination of FIB-4 and the other fibrosis indices to predict the presence of high-risk varices and their sensitivity, specificity, PPV and NPV in the estimation cohort are shown in Table 4. FIB-4 alone and a combination of FIB-4 and the other fibrosis indices showed a sensitivity of 100% and NPV of 100%, indicating that FIB-4 alone is able to identify 100% of patients with cirrhosis, thus enabling the potential prevention of unnecessary screening endoscopy.

### 3.6. Performance of Other Fibrosis Indices

The diagnostic performances of FIB-4 the other fibrosis indices were evaluated using ROC curves in the validation cohort. Using FIB-4, we could rule out the presence of any varices in 21.3% of patients with a miss rate of 0%. HA, PLT, ELF score, 7S collagen, APRI, M2BPGi, PIIINP and TIMP-1 enabled us to rule out the presence of any varices in 15.9%, 45.7%, 50.4%, 34.6%, 33.9%, 26.8%, 7.9% and 54.3% patients, respectively, with a miss rate of 5.3%, 3.5%, 7.8%, 4.5%, 4.7%, 5.9%, 6.3% and 7.2%, respectively (Table 5).

### 3.7. Univariate and Multivariate Analyses of Variables Associated with High-Risk Varices

Univariate and multivariate analysis demonstrated that no fibrosis index was found to be an independent variable for predicting high-risk varices (Appendix A).

## 4. Discussion

The present retrospective study, including 478 patients with cirrhosis who underwent esophagogastroduodenoscopy, revealed that FIB-4 correctly stratified patients with cirrhosis without high-risk varices. To the best of our knowledge, this is the first study to examine serum fibrosis indices for the exclusion of high-risk varices and to identify patients who could potentially avoid endoscopy. The Baveno IV consensus indicates the usefulness of liver stiffness in combination with platelet count for the identification of patients without high-risk varices. However, due to the limited availability of transient elastography, there is increasing interest in the development of simple serum biomarkers that can distinguish patients without high-risk varices to reduce the burden of endoscopic variceal surveillance.

The main objective of the present study was to determine the role of serum fibrosis indices in excluding the presence of high-risk varices in patients with cirrhosis. The Baveno VI criteria have been shown to potentially avoid 15–25% of variceal screening endoscopies, with a rate of 2% for missed varices requiring treatment [22,23]. These criteria are presumed to be accurate, with a minimum number of incorrect negative predictions [5]. Otherwise, prevention of variceal bleeding is likely to be referred in an incorrect classification. In the current study, 21.3% (27/127) and 14.8% (52/351) of the patients in the estimation and validation cohorts, respectively, avoided esophagogastroduodenoscopy, and there were no missed high-risk varices in either cohort. The observed differences in the rate of avoidable esophagogastroduodenoscopy between the estimation and validation cohorts might be explained by the significantly higher proportion of patients with high-risk varices in the estimation cohort compared with the validation cohort. The number of patients with portal hypertension having false-positive high-risk varices might increase with the progression of hepatic fibrosis in patients without clinically significant portal hypertension [24,25]. These findings reinforce the fact that the rate of potentially avoidable screening endoscopy is decreased with the progression of liver cirrhosis and esophageal varices.

We demonstrated that the use of a single fibrosis index, FIB-4, completely avoided unnecessary endoscopies. Combination of FIB-4 with any other fibrosis indices could achieve NPV of 100% because the NPV of an FIB4 cutoff value of 2.78 for high-risk varices was 100%. Combination of FIB-4 with FibroScan^®^ was reported to increase the sensitivity and NPV compared to either test alone [26]. However, no improvement was observed in the rate of false positivity in the internal validation cohort. Although unclear, differences in index cutoff values, etiological background and portal hypertension parameters among the studies may have contributed to the observed differences in the diagnostic performance of FIB-4 in combination with fibrosis indices to estimate high-risk varices. Moreover, a FIB-4 cutoff of 3.2, while avoiding a larger number of endoscopies, may also increase the odds of missing high-risk varices. These results further support the hypothesis that FIB-4 alone can adequately identify patients without high-risk varices. However, univariate and multivariate analyses demonstrated that no factor was found to be an independent variable for predicting high risky varices. Larger studies are necessary to elucidate the relationship between serum fibrosis indices and portal hypertension in patients with cirrhosis. Nevertheless, we have shown that patients with cirrhosis can safely avoid esophagogastroduodenoscopy by utilizing FIB-4, which comprises simple clinical parameters and not imaging modalities. Specifically, patients with cirrhosis and a FIB-4 of <2.78 may be unlikely to have high-risk varices. We propose that patients with cirrhosis and a FIB-4 of <2.78 should undergo FIB-4 reassessment every 6–12 months concurrently with the other blood tests until the index value reaches 2.78 (Figure 4) and that those with a FIB-4 of ≥2.78 should undergo variceal screening endoscopy.

Several limitations of the present study should be acknowledged. First, this was a retrospective, single-center, observational study, including a small number of patients with cirrhosis (statistical power: 0.9). Second, approximately 15% of the patients in the validation cohorts avoided esophagogastroduodenoscopy. Third, the patients did not undergo transient elastography measurements for a direct performance comparison between the Baveno VI criteria and FIB-4 to potentially avoid variceal screening endoscopy. Fourth, although no differences in age were observed between two cohorts, FIB4 uses four variables, including age; appropriate cutoff points may differ between age groups. Further studies should evaluate the diagnostic accuracy of FIB-4 for the absence of high-risk varices in patients with liver cirrhosis.

In conclusion, a FIB-4 of <2.78 correctly identified 100% of patients with cirrhosis who could potentially avoid endoscopy. FIB-4, which can be determined repeatedly using simple means, fulfills the requisites of an accurate index that can replace screening esophagogastroduodenoscopy for primary care physicians. Longitudinal data collection will facilitate the adoption of these recommendations.

## Figures and Tables

**Figure 1 jcm-09-03510-f001:**
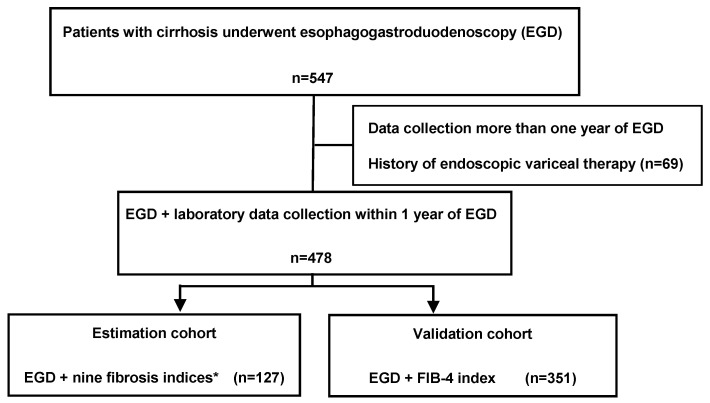
Flow chart showing the division of patients into the estimation and validation cohorts. A total of 547 patients were identified in the database of patients with cirrhosis treated during the study period. After the exclusion of patients with cirrhosis who did not undergo laboratory tests within one year of esophagogastroduodenoscopy (EGD) and those with a history of endoscopic variceal therapy, the remaining 478 patients were divided into the estimation and validation groups. The estimation cohort comprised patients with data available for the following nine fibrosis indices within one year of EGD: platelet count, hyaluronic acid, 7S fragment of type 4 collagen, procollagen type III N-terminal peptide, tissue inhibitor of metalloproteinase-1, Mac-2-binding protein glycosylation isomer, fibrosis index based on four factors (FIB-4), aspartate aminotransferase/platelet ratio index and enhanced liver fibrosis score. The validation cohort comprised patients who underwent EGD with data available for FIB-4 index within one year of EGD. * the fibrosis index based on four factors (Fib-4 index); enhances liver fibrosis score (ELF score); 7S fragment of type 4 collagen (7S collagen); the aspartate aminotransferase-to-platelet ratio index (APRI); mac-2-binding protein glycosylation isomer (M2BPGi); type 3 procollagen-N-peptide (PIIINP); tissue inhibitor of metalloproteinase 1 (TIMP-1), Platelet, Hyaluronic acid (HA).

**Figure 2 jcm-09-03510-f002:**
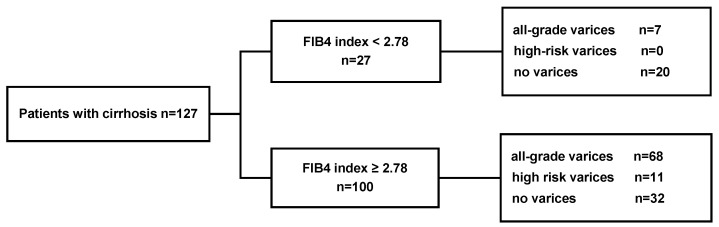
Rates of all-grade varices and high-risk varices according to a FIB-4 cutoff of 2.78 in the estimation cohort of patients with cirrhosis. EV, esophagealvarices; FIB-4, fibrosis index based on four factors.

**Figure 3 jcm-09-03510-f003:**
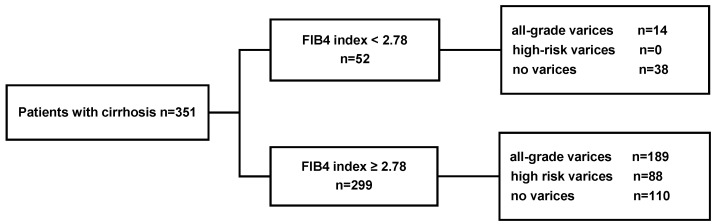
Rates of all-grade varices and high-risk varices according to a FIB-4 cutoff of 2.78 in the validation cohort of patients with cirrhosis. EV, esophagealvarices; FIB-4, fibrosis index based on four factors.

**Figure 4 jcm-09-03510-f004:**
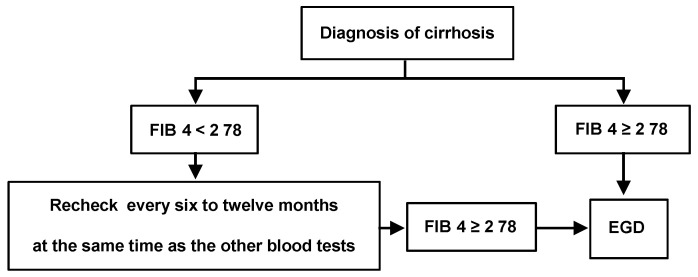
Algorithm for screening and follow-up of esophageal varices. The flowchart shows the algorithm for the management of esophageal varices using FIB-4. Patients with a FIB-4 of ≥2.78 should undergo screening with EGD. Determination of FIB-4 every 6–12 months concurrently with the other blood tests is recommended for those with a FIB-4 of <2.78 until the FIB-4 reaches 2.78. EGD, esophagogastroduodenoscopy; EV, esophagealvarices; FIB-4, fibrosis index based on four factors.

**Table 1 jcm-09-03510-t001:** Clinical and demographic characteristics of patients with cirrhosis.

	Estimation Cohort (*n* = 127)	Validation Cohort (*n* = 351)	*p* Value
**Gender (male/female)**	82/45	217/134	0.67
**Etiology (HBV/HCV/alcohol/NASH/other)**	13/50/35/12/17	42/131/92/30/56	0.52
**Child-pugh classification (A/B/C)**	83/37/7	202/122/27	0.29
**Age (years) ^a^**	71 (63–76.5)	69 (62.5–76)	0.42
**AST (IU/L) ^a^**	40 (28–54.5)	39 (28–55)	0.81
**ALT (IU/L) ^a^**	28 (19–43)	27 (18–43)	0.51
**ALB (g/dL) ^a^**	3.9 (3.2–4.3)	3.6 (3.1–4.1)	0.01
**Total Bilirubin (mg/dl) ^a^**	1.68 (0.8–1.68)	1.0 (0.7–1.4)	0.21
**Prothrombin time (%) ^a^**	74 (63.5–85)	75 (64–87)	0.32
**Platelet (10^3^/μL) ^a^**	11 (8.25–14.5)	10.1 (7.1–14.3)	0.09
**Model for End-Stage Liver Disease (MELD) scores ^a^**	8 (5–11.5)	7 (4–10.3)	0.47
**Hyaluronic acid (ng/mL) ^a^**	402 (172.2–724.6)	-	
**7S collagen (ng/mL) ^a^**	7.6 (5.25–11.05)	-	
**PIIINP (ng/mL) ^a^**	0.9 (0.6–1.1)	-	
**TIMP-1 (ng/mL) ^a^**	348.1 (269.2–486.3)	-	
**M2BPGi (COI) ^a^**	3.18 (1.36–6.56)	-	
**FIB-4 index ^a^**	4.54 (3.00–7.32)	5.2 (3.5–8.2)	0.065
**APRI ^a^**	1.16 (0.80–1.86)	1.37 (0.77–2.28)	0.29
**ELF score ^a^**	11.7 (10.8–12.6)	-	
**The form (F) of esophageal varices F0/F1/F2-F3**	52/64/11	150/115/86	*p <* 0.001

AST, aspartate aminotransferase; ALT, alanine aminotransferase; ALB, albumin; 7S collagen, 7S fragment of type 4 collagen; PIIINP, type 3 procollagen-N-peptide; TIMP-1, Tissue Inhibitor of Metalloproteinase 1; M2BPGi, Mac-2-binding protein glycosylation isomer; FIB-4, the fibrosis index based on four factors; APRI, the aspartate aminotransferase-to-platelet ratio index; ELF, Enhances Liver Fibrosis. ^a^, Mean ± standard error of mean.

**Table 2 jcm-09-03510-t002:** Performance of fibrosis indices for the prediction of all-grade varices and high-risk varices in the estimation cohort (*n* = 127).

Fibrosis Indices	All-Grade Varices						High-Risk Varices				
	Cut-Off	AUROC (95%CI)	Se	Sp	PPV	NPV	AUROC (95%CI)	Se	Sp	PPV	NPV
**FIB4-index**	2.78	0.69 (0.60–0.78)	0.91	0.39	0.68	0.74	0.53 (0.40–0.66)	1.00	0.28	0.12	1.00
**Hyaluronic acid (ng/mL)**	110.63	0.63 (0.53–0.73)	0.95	0.29	0.66	0.79	0.50 (0.34–0.66)	0.91	0.28	0.11	0.97
**Platelet (×10^3^/μL)**	11.9	0.63 (0.61–0.79)	0.69	0.65	0.74	0.60	0.56 (0.45–0.74)	0.82	0.47	0.13	0.97
**ELF score**	11.75	0.63 (0.53–0.73)	0.60	0.65	0.71	0.53	0.48 (0.33–0.64)	0.91	0.21	0.10	0.96
**7S collagen (ng/mL)**	6.1	0.66 (0.56–0.75)	0.75	0.50	0.68	0.58	0.53 (0.39–0.67)	0.82	0.36	0.11	0.96
**APRI**	0.89	0.66 (0.56–0.75)	0.60	0.65	0.71	0.53	0.56 (0.42–0.70)	0.73	0.53	0.13	0.95
**M2BPGi (COI)**	1.47	0.63 (0.53–0.73)	0.85	0.44	0.69	0.68	0.53 (0.38–0.69)	0.73	0.44	0.11	0.94
**PIIINP (ng/mL)**	0.60	0.54 (0.44–0.65)	0.92	0.19	1.00	0.43	0.48 (0.31–0.66)	0.09	1.00	1.00	0.92
**TIMP-1 (ng/mL)**	379.9	0.56 (0.48–0.68)	0.52	0.65	0.67	0.50	0.48 (0.31–0.66)	0.46	0.55	0.07	0.82

The diagnostic accuracy of the fibrosis indices for prediction of high risky varices was evaluated using the optimal cutoff values for all grade varices. Se, sensitivity; Sp, specificity; PPV, positive predictive value; NPV, negative predictive value; FIB-4 index, the fibrosis index based on four factors; ELF score, enhances liver fibrosis score; 7S collagen, 7S fragment of type 4 collagen; APRI, the aspartate aminotransferase-to-platelet ratio index; M2BPGi, mac-2-binding protein glycosylation isomer; PIIINP, type 3 procollagen-N-peptide; TIMP-1, tissue inhibitor of metalloproteinase 1.

**Table 3 jcm-09-03510-t003:** Performance of FIB-4, platelet and APRI in the prediction of high-risk varices in the validation cohort (*n* = 351).

Validation Set (*n* = 351)	High-Risk Varices			
	Se	Sp	PPV	NPV
**FIB 4-index cut-off 2.78**	1.00	0.20	0.29	1.00
**Platelet cut-off 11.9**	0.13	0.55	0.09	0.65
**APRI cut-off 0.89**	0.91	0.37	0.33	0.93

FIB-4 index, the fibrosis index based on four factors; APRI, the aspartate aminotransferase-to-platelet ratio index; Se, sensitivity; Sp, specificity; PPV, positive predictive value; NPV, negative predictive value.

**Table 4 jcm-09-03510-t004:** Performance of a combination of FIB-4 and the other fibrosis indices for the prediction of high-risk varices in the estimation cohort (*n* = 127).

A Combination of FIB-4 and the other Fibrosis Indices	Se	Sp	PPV	NPV
Fib4 cut-off 2.78	1.0	0.284	0.117	1.0
Fib4 < 2.78 and ELF < 11.75	1.0	0.172	0.103	1.0
Fib4 < 2.78 and M2 < 1.47	1.0	0.155	0.101	1.0
Fib4 < 2.78 and ELF < 11.75 and M2BPGi	1.0	0.147	0.1	1.0
Fib4 < 2.78 and HA < 110.63	1.0	0.103	0.096	1.0
Fib4 < 2.78 and 7S collagen < 6.1	1.0	0.164	0.102	1.0
Fib4 < 2.78 and P3P < 0.6	1.0	0.078	0.093	1.0
Fib4 < 2.78 and 7S collagen < 6.1 and HA < 110.63	1.0	0.103	0.096	1.0
Fib4 < 2.78 and 7S collagen < 6.1 and M2 < 1.47	1.0	0.155	0.0101	1.0
Fib4 < 2.78 and ELF and 7S collagen < 6.1	1.0	0.147	0.1	1.0
Fib4 < 2.78 and P3P < 0.6 and HA < 110.63	1.0	0.052	0.091	1.0
Fib4 < 2.78 and P3P < 0.6 and M2 < 1.47	1.0	0.078	0.093	1.0
Fib4 < 2.78 and P3P < 0.6 and 7S collagen < 6.1	1.0	0.078	0.093	1.0
Fib4 < 2.776 and M2 < 1.47 and HA < 110.63	1.0	0.103	0.093	1.0

Se, sensitivity; Sp, specificity; PPV, positive predictive value; NPV, negative predictive value; FIB-4 index, the fibrosis index based on four factors; ELF score, enhances liver fibrosis score; 7S collagen, 7S fragment of type 4 collagen; APRI, the aspartate aminotransferase-to-platelet ratio index; M2BPGi, mac-2-binding protein glycosylation isomer; PIIINP, type 3 procollagen-N-peptide; TIMP-1,tissue inhibitor of metalloproteinase 1.

**Table 5 jcm-09-03510-t005:** Rate of ruling out and missing high-risk varices in the estimation cohort (*n* = 127).

	Cut-Off	Rate of Ruling Out High-Risk Varices	Rate of Missing High-Risk Varices
**FIB-4 index**	2.78	27 (21.3%)	0 (0%)
**Hyaluronic acid** (ng/mL)	110.63	19 (15.9%)	1 (5.3%)
**Platelet** (×10^3^/μL)	11.9	58 (45.7%)	2 (3.5%)
**ELF score**	11.75	64 (50.4%)	5 (7.8%)
**7S collagen** (ng/mL)	6.1	44 (34.6%)	2 (4.5%)
**APRI**	0.89	43 (33.9%)	2 (4.7%)
**M2BPGi** (COI)	1.47	34 (26.8%)	2 (5.9%)
**PIIINP** (ng/mL)	0.60	16 (7.9%)	1 (6.3%)
**TIMP-1** (ng/mL)	379.9	69 (54.3%)	5 (7.2%)

7S collagen, 7S fragment of type 4 collagen; PIIINP, type 3 procollagen-N-peptide; TIMP-1, Tissue Inhibitor of Metalloproteinase 1; M2BPGi, Mac-2-binding protein glycosylation isomer; FIB-4, the fibrosis index based on four factors; APRI, the aspartate aminotransferase-to-platelet ratio index; ELF, Enhances Liver Fibrosis.

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
