# Peer review of "Accuracy of Fibrosis-4 Index in Identification of Patients with Cirrhosis Who Could Potentially Avoid Variceal Screening Endoscopy"

_jcm, 2020, doi:10.3390/jcm9113510_

Round 1
Reviewer 1 Report
No further comments
Reviewer 2 Report
Thank you for this contribution to improving the diagnosis of fibrosis.
This manuscript is a resubmission of an earlier submission. The following is a list of the peer review reports and author responses from that submission.
Round 1
Reviewer 1 Report
I thank the Authors for their considered responses to the comments I made. The article is much improved.
Author Response
N/A
Reviewer 2 Report
This is an interesting and well written paper. However, some issuses should be addressed by the authors
1) How did they conclude that FIB4 should be repeated every 6 months for those with <2.78?
2) 2nd paragraph in the results. What did the authors mean regarding "the pts in the validation cohort developed significantly more high risk varices..."? During follow up?
Reviewer 3 Report
Two spelling mistakes:
Risky
Forth
Author Response
N/A
Reviewer 4 Report
The paper by Ishida et al. "Accuracy of fibrosis-4 index in identification of patients with cirrhosis who may possibly avoid variceal screening endoscopy" is a good paper, write well.
the principal limitations, as reported, are the small number of paients (for this kind of the research the statistical power it's important and must be decleared).
second the patients did not undergo transient elastography measurements and this is an important limitations for the study.
15% of patients in validation cohort had FIB4<2.78. Further studies should evaluate the diagnostic accuracy of FIB-4 for the absence of high risk varices in patients with liver cirrhosis.
Author Response
September 21, 2020
Journal of Clinical Medicine
Manuscript ID: jcm-844312
Title: Accuracy of fibrosis-4 index in identification of patients with cirrhosis who could potentially avoid variceal screening endoscopy
Authors: Koji Ishida, Tadashi Namisaki, Koji Murata, Yuki Fujimoto, Souichi Takeda, Masahide Enomoto, Hiroyuki Ogawa, Hirotetsu Takagi, Yuki Tsuji, Daisuke Kaya, Takahiro Ozutsumi, Yukihisa Fujinaga, Masanori Furukawa, Yasuhiko Sawada, Koh Kitagawa, Shinya Sato, Norihisa Nishimura, Hiroaki Takaya, Kosuke Kaji, Naotaka Shimozato, Hideto Kawaratani, Kei Moriya, Takemi Akahane, Akira Mitoro, Hitoshi Yoshiji
We thank you and the reviewers for your helpful comments and suggestions about our manuscript. We have thoroughly addressed all the concerns and issues that you have raised and have accordingly revised our manuscript. All the changes in the revised manuscript are highlighted in yellow. We believe that the manuscript is considerably improved and trust that it is now suitable for publication in the Journal of Clinical Medicine. Once again, we acknowledge your comments that have been extremely valuable in helping us improve the quality of our manuscript. We have provided point-by-point responses to the reviewers’ comments below.
Reviewer: 3
The paper by Ishida et al. "Accuracy of fibrosis-4 index in identification of patients with cirrhosis who may possibly avoid variceal screening endoscopy" is a good paper, write well.
the principal limitations, as reported, are the small number of paients (for this kind of the research the statistical power it's important and must be decleared).
second the patients did not undergo transient elastography measurements and this is an important limitations for the study.
15% of patients in validation cohort had FIB4<2.78. Further studies should evaluate the diagnostic accuracy of FIB-4 for the absence of high risk varices in patients with liver cirrhosis.
Author response
We agree with your valuable comment. First, this was a retrospective, single-centre, observational study including a small number of patients with cirrhosis(statistical power:0.9). Second, approximately 15% of the patients in the validation cohorts avoided esophagogastroduodenoscopy. Third, the patients did not undergo transient elastography measurements for a direct performance comparison between the Baveno VI criteria and FIB-4 to potentially avoid variceal screening endoscopy. Fourth, although no differences in age were observed between two cohorts, FIB4 uses four variables, including age, appropriate cutoff points may differ between age groups. Further studies should evaluate the diagnostic accuracy of FIB-4 for the absence of high risk varices in patients with liver cirrhosis. We have described these limitations in the revised manuscript on page 10 lines 8–16.
Round 2
Reviewer 4 Report
thanks for your response